
**Italian Tsunami Effects Database (ITED): the first database of tsunami effects observed along the Italian coasts**

Alessandra Maramai [1], Laura Graziani [1], Beatriz Brizuela [1]

[1] Istituto Nazionale di Geofisica e Vulcanologia, Rome, 00143, Italy

*Correspondence to*: Laura Graziani (laura.graziani@ingv.it)

**Abstract.** The Italian Tsunami Effects Database (ITED), consists of an ensemble of records reporting tsunami effects observed at several Observation Points (OP) along the Italian coasts from historical times. ITED was compiled starting from the Euro Mediterranean Tsunami Catalogue (EMTC) and it focuses on the propagation effects observed along the Italian coasts, providing information on how each locality was interested by tsunami effects over time. The effects reported in ITED are related to

tsunamis occurred within the Italian territory and contained in the EMTC; these events were excerpt, analyzed and updated according to recent studies published in literature.

The database can be accessed through a web GIS application, that displays the location of the OPs indicating for each of them the description of tsunami effects found in literature and the corresponding bibliographic references as well as the metrics related to the observed event. Based on those

descriptions, the estimate value of the tsunami intensity has been assigned to each OP, according to both the Sieberg-Ambraseys and the Papadopoulos-Imamura scales. All the ITED data, including quantitative data such as runup, inundation, withdrawal, can be retrieved by accessing online the database through the WebApp that was expressly designed and built for this purpose.

ITED contains 300 observations of tsunami effects at 225 OPs referred to 186 Italian main localities,

hereafter called Place Name (PN) related to 72 Italian tsunamis. The database provides also the tsunami-history for each PN, allowing the end user to have a complete picture of how the PN is prone to tsunami effects. The realization of ITED was also the occasion to update Italian tsunamis contained in the EMTC, leading to the release of a new version of the EMTC catalogue, named EMTC2.0.

ITED was specifically built to meet the needs of the tsunami hazard community, thus providing useful

information that can improve the knowledge on how much the national territory is exposed to tsunami risk.

Keywords: tsunami, tsunami effects, historical tsunami, Italian coasts, tsunami intensity, tsunami history




# 1 Introduction

The knowledge of past tsunamis, like for the other natural phenomena, is essential for the comprehension of the phenomenon itself and for developing hazard models and scenarios that could increase the awareness in order to be prepared for future events. Examining historical records is a

fundamental tool for these purposes as the realization of reliable and complete catalogues represents the first step for evaluating the tsunami hazard assessments of a region and for risk mitigation. In fact, they also represent the starting point for all activities aimed at increasing public awareness and, therefore, at reducing the impact of tsunamis.

In the last decade, the growing interest in focusing on tsunami studies in the European region led to the

compilation of the Euro-Mediterranean Tsunamis Catalogue (EMTC, Maramai et al., 2014). EMTC is a unified tsunami catalogue containing 290 tsunamis generated in the European and Mediterranean seas since 6150 B.C. to 2014. It is the result of a detailed review of all the regional catalogues available in literature covering the study area and is a collection of tsunamis that, starting from the generating cause (earthquakes, volcanic activity, landslides), provides parameters and information on the event as a

whole. The parameters related to each tsunami are contained in a descriptive sheet that reports all the information of the event (date, region, reliability, tsunami intensity, maximum run-up) and of the generating cause (i.e. geographical coordinates, earthquake magnitude, earthquake intensity, etc.).

According to EMTC entries, Italy is one of the Euro-Mediterranean countries most prone to tsunami threat, as confirmed by the probabilistic tsunami hazard assessment (PTHA) recently performed for the

Euro-Mediterranean region (http://ai2lab.org/tsumapsneam/interactive-hazard-curve-tool/). The Italian section of EMTC is indeed one of the richest sections of the catalogue counting for 70 tsunamis from the 79 A.D. Vesuvius eruption to the 2002 Stromboli (Aeolian Islands) tsunami. The majority of the EMTC tsunamis are ranked as high quality being well documented. The quality of each event is given by the *reliability*, an index ranging from 0 (very improbable tsunami) to 4 (definite tsunami). Figure 1

summarizes the number of Italian tsunamis according to their reliability.

The Italian Tsunami Effects Database (ITED) is an ancillary database that provides punctual and detailed information on the effects of tsunamis observed on the sites; unlike EMTC, which focuses on the source of a tsunami, it centers on the propagation effects of tsunami. ITED was compiled starting from the general descriptions of the Italian tsunamis present in the EMTC and for its realization an

accurate research and analysis of the recent studies available in literature regarding Italian events was carried out, in order to include new information on the effects observed and/or measured at different locations. This process enabled us to update also the Italian tsunamis contained in the EMTC, leading to the release of a new version of the EMTC, named EMTC2.0.

The two databases, ITED and EMTC2.0, are strictly interconnected and to better exploit data contained

therein a GIS WebApp was built for allowing the user to retrieve information, to switch from one database to another and to download data.



## 2 ITED data and EMTC2.0

According to EMTC, Italian coasts were struck by 70 tsunamis, starting from 79 A.D. to 2002. Moderate magnitude earthquakes that occurred offshore or inland close to the coastline mainly

generated Italian tsunamis. Volcanic activity of Stromboli (Aeolian Islands) and Vesuvius volcanoes triggered about ten tsunamis, mostly with local effects. On the basis of the observed effects most of tsunamis can be classified as low to rather strong intensity events, but a few of them have caused severe damage and victims and they have been considered as high intensity events (Fig. 2).

The analysis of the catalogue highlighted that Southern Italy and in particular Eastern Sicily and

Southern Calabria coasts are the most severely affected segments along the Italian peninsula.

ITED consists of an ensemble of records reporting tsunami effects observed/measured at several Observation Points (OP) along the Italian coasts from historical times to the present. It was compiled by excerpting and analyzing the information already available in the portion of the EMTC dataset related to the Italian tsunamis, aiming to enhance the usability of the data. ITED OPs are geographical

points in which tsunami effects were observed and they were associated to the coordinates of the main localities contained in the INGV Gazzetter, hereafter called Place Names (PNs).

In Fig. 3 the main screen of ITED WebApp is visible, showing the geographical distribution of the PNs.

When the descriptions of the tsunami were imprecise not allowing an exact identification of the OP, the

coordinates of the nearest PN were assigned. In some cases, usually when post event survey measures were available, descriptions of effects were so detailed that for a PN more than one OP is identifiable. In Fig. 4 the example of Stromboli Island (Aeolian Islands) OPs and PN is shown.

In ITED for each OP the description of the tsunami effects found in literature is given, accompanied by the corresponding bibliographic references and, according to the description, the estimate of the

intensity of the phenomenon at the site (Local Intensity) is also provided. Intensity was assessed according to both the Sieberg-Ambraseys (Ambraseys, 1962) and the Papadopoulos-Imamura (Papadopoulos and Imamura, 2001) scales (Fig. 4).

All quantitative data contained in the description of the effects and visible in the pop-up (runup, inundation, withdrawal, etc. as shown in Fig. 4) populates an attribute table that can be retrieved by

querying the database.

In order to improve the quality of the data, all the Italian tsunamis of seismic origin have been linked to the Parametric Catalogue of Italian Earthquakes (CPTI15, Rovida et al., 2016), adopted as the reference earthquake catalogue. This choice leads us to reappraise some Italian events that in the EMTC had a not reliable seismic cause. ITED is interoperable with CPTI15 and the Italian Archive of Historical

Earthquake Data (ASMI- Rovida et al., Eds.) database. The interoperability is possible by linking the earthquake identifier as assigned by CPTI15. Moreover the INGV Gazzetter adopted in ITED is the



unified geographical reference system for macroseismic observations of the Italian Macroseismic Database (DBMI - Locati et al., 2016).

In ITED new data were added following a cross-checking with the recent published Catalogue of Strong Earthquakes in Italy (CFTI5Med, Guidoboni et al., 2018), in which, when the earthquake

triggered a tsunami, the documentation of the event is also accompanied by the description of the tsunami effects.

ITED contains 300 observations of tsunami effects at 225 OPs, providing quantitative and descriptive information. Among the 186 PNs present in the database 49 experienced tsunami effects more than once. For each of them details of each observation is supplied allowing the user to retrieve the tsunami-

history of the place, having a complete picture of how the PN is prone to tsunami effect. For ITED realization all Italian tsunamis have been taken into account, apart from the few events generated outside national boundaries that may have caused tsunami effects in the Italian coast.

With regard to the observations contained in ITED, the December 28, 1908 Messina tsunami is by far the most documented event in Italy having 115 OPs along the Italian coasts. They mostly come from

the reports of Platania (Platania, 1909) and Baratta (Baratta, 1910) that, soon after the event, performed independently detailed surveys along Sicily and Calabria coasts. The 1908 tsunami spread over great distances reaching the island of Malta to the south (about 250 km away) with a sea level rise of more than one meter causing slight damage. To the north it was recorded at the tide gauge of Civitavecchia, located about 450 km away. In Fig. 5 a snapshot of ITED WepApp that shows the OPs for the 1908

Messina tsunami.

As far as runup values are concerned, in ITED there are 91 OPs for which a runup value is specified: 7 OPs show values higher than 10 metres, all generated by the 1908 Messina and by the 2002 Stromboli (Aeolian Island) events.

The 91 OPs for which runup value is available were generated by 7 tsunamis, most of which of seismic

origin, with the exception of one event associated with a large landslide triggered by an earthquake (February 6, 1783) and of two events associated to the Stromboli volcano activity (1930 and 2002).

The 6 February, 1783 tsunami was generated by a $6 \times 10^6$ m$^3$ landslide (Zaniboni et al. 2019, Zaniboni et al. 2016) and caused more than 1500 fatalities at Scilla (Tyrrhenian Calabria) where most of the people, frightened by the ongoing sequence of earthquakes, sought on the beach close to the town and were

surprised by the unexpected waves that reached the roofs of the buildings with a runup of 9 meters (Graziani et al. 2006).

ITED contains 70 OPs where inundation values are specified, some of them are the same as those for which the runup values are also available. For 32 OPs the inundation extent is greater or equal to 100 meters. According to coeval sources, the maximum value was observed during the January 11, 1693

tsunami in Eastern Sicily where: *"At Mascali the sea flooded the shore for about 1 mile (1,4 Km) inland"* (Boccone, 1697).


As mentioned before, ITED allows the user to read the tsunami history of all the PNs reported, and know how many times a locality has been hit by a tsunami and how severe was the impact. Among the 49 OPs that experienced tsunami effects more than once, Messina, Catania Stromboli and Scilla are the places with the most remarkable tsunami histories, being affected during times by five or more tsunami events. The fact that all these towns are located between Sicily and Calabria highlights once again that these are the Italian areas most exposed to the tsunami. In Fig. 7 the tsunami-histories of Catania, Messina, Stromboli and Scilla are shown. It can be noticed that the tsunami history of Stromboli is populated by 8 tsunamis corresponding to 17 observations starting from 1900, suggesting the lack of documentation regarding tsunami effects occurred before that date.

Concerning the impact of the Italian tsunamis in terms of geographical area involved, 14 Italian tsunamis were observed at regional scale that is for a distance greater than 100 km from the source. Besides the 1908 Messina tsunami, it is worth mentioning the 11 January 1693 Eastern Sicily that was observed along the whole Ionian coasts of Sicily from Siracusa to Messina and the 1887 Ligurian tsunami that involved more than 200 kilometers of coast, from Cannes (France) to the Rapallo gulf (Liguria) with runup of more than 1m and remarkable sea withdrawals leaving boats stranded in many localities.

All the regional events were generated by earthquakes, with the exception of the 2002 Stromboli Island tsunami, which was observed up to the coasts of Campania (about 140 km North). This event was triggered by huge submarine ($20\times10^6$ m$^3$, Tinti et al., 2005) and sub-aereal ($4$-$9\times10^6$ m$^3$, Tinti et al., 2005) slides occurred during a relevant volcanic eruption on the Sciara del Fuoco, the steep flank of the Stromboli volcano. In the northern part of the island a maximum runup of 10.90 meters was measured (Tinti et al. 2006).

The realization of ITED was also the occasion to update Italian tsunamis contained in the EMTC, leading to the realization of a new release, called EMTC2.0. Compared to the old version, EMTC2.0 has undergone several changes. First of all, two new events have been added so that the number of Italian tsunamis increases from 70 to 72, more information enriched the knowledge on the tsunamis present in the catalogue and, accordingly to the new achieved info, some tsunami parameters, such as reliability, intensity, generating cause, etc. have changed. Due to these relevant amendments, 63 Italian events were updated with respect to EMTC.

As regards the two new events inserted in the database, these are the December 10, 1542 (Eastern Sicily) and the December 23, 1690 (Central Adriatic). The first one, known as the Val di Noto earthquake, is a destructive shock (M=6.7, Rovida et al., 2016) occurred during a seismic sequence in Southeastern Sicily, causing the destruction of some villages. Coeval sources referred that *"....after the shock the city of Augusta was almost submerged by the sea and many people were drowned"* (Guidoboni et al., 2018). About the 1690 event, a severe earthquake (M=5.6, Rovida et al., 2016) hit the Central Adriatic region provoking victims and severe damage at Ancona and in some neighboring





villages. According to the coeval source Bonito (1691), *"in the beach of Ancona the boats touched the sea bottom and then they lifted up being shaken due to the sea water agitation."*

In addition to the inclusion of the two new tsunamis, EMTC2.0 presents a number of differences in some tsunami parameters respect to EMTC. Table 1 shows the number of events for which parameters

were updated in EMTC2.0.

## 3 ITED WebApp

The information contained in ITED has been made available to the public through the display of a web application that allows the users to visualize the geo-referenced information on a map. The dedicated

WebApp has been developed using the Esri ArcGIS online environment and can be freely accessed without the need of an Esri user account. The ITED WebApp, accessible through https://tsunamiarchive.ingv.it/ited , hosts five layers with information regarding the database with all the tsunami effects observed at the Italian Coast (ITED). The layers contained in the WebApp are: i) ITED_Observation_Point, where the OP are visible and colored according to the tsunami intensity, ii)

ITED_Place Name with the main struck localities complemented by the tsunami histories, iii) ITED_inundation and iiii) ITED_runup where the OPs with reported inundation and runup are visible with symbols proportional to the extent of the metric and the iiiii) EMTC2.0 layer that hosts the new release of database of the tsunami events. Figure 7 shows a screenshot of the WebApp with the ITED_Observation Point layer and the list of all the layers is also visible on the screen.

The ITED_Observation Point layer contains 300 observations reported in 225 OPs where tsunami effects have been noticed, indicating general information regarding the tsunami event, specific information observed at the OP (Local tsunami intensity) and a short description. The information is displayed on the ITED WebApp as a point layer, where the color represents the intensity value according to the Sieberg Ambraseys scale (S-A) assigned to each OP. This information can be accessed

through a pop-up window, by clicking in each OP (Fig. 5)

The ITED_Place Name layer directs to the tsunami history of all the PNs: a chart showing all the observed events at OPs associated to the specific PN and a table reporting date, intensity and a short description of the effects is visible on the pop-up (Fig. 8).

As already mentioned, ITED WebApp also hosts two layers, ITED_inundation and ITED_runup, that

display these metrics at each OP indicating the parameter value and the tsunami event related to the observation (Fig. 9).

As mentioned before, the ITED database is deeply related to the EMTC2.0 catalogue, which is also displayed in ITED WebApp as a point layer called EMTC2.0. The events compiled on EMTC2.0 are divided into 4 main causes (earthquakes, landslides, volcanic eruptions and unknown cause) and the

size of the symbol represents the S-A tsunami intensity. As it was in the old version, information such


as date, time, reliability, cause, intensity, magnitude (if seismic), coordinates, macroseismic intensity and a detailed tsunami description including the related bibliographical sources can be accessed by clicking on each point of the layer (Fig. 10)

The ITED WebApp consents the user to customize which kind information to retrieve; through five

widgets the user can choose the layers to be displayed (layer list), can query each of the available layers (query) (Fig. 11), filter information (group filter), and switch the view from EMTC2.0 to ITED (swipe). The tabular information contained in each layer of the WebApp can be exported in csv format by using several filtering expressions (i.e., selecting data from the extent viewed, or by date, reliability, cause, locality etc.; or by a combination of several of these parameters). In addition, the "Print" widget allows

the user to export and print maps with the results of their specific queries or a general view of the database.

## 4 Conclusions

The ITED database was built on the basis of information contained in EMTC relating to observations

of the effects of tsunamis along the Italian coasts. ITED has been developed using the Esri ArcGIS online environment and a dedicated WebApp has been created that allows the user to retrieve information, to switch from ITED to EMTC2.0 and vice versa and to download data.

The implementation of ITED leads to a new release of EMTC, called EMTC2.0, which acquires ITED data for the Italian section. Compared to EMTC, EMTC2.0 includes two new Italian tsunamis, new

information has been added and some tsunami parameters, such as the reliability, intensity, generating cause, etc. have changed.

ITED was specifically built to meet the needs of the tsunami hazard community, thus providing useful information that can improve the knowledge of how much the Italian coasts are exposed to tsunami attack. The database contains a great number of reliable data that can contribute to validate hazard

assessment along the Italian coasts and can give useful indications for the realization of inundation maps and emergency plans. ITED can also be used as a tool for education activities aimed at increasing public awareness and, therefore, at reducing the hazard of tsunamis.

To complete the knowledge of how Italian coast were hit by historical tsunamis it is important to take into account also distant sources, including in ITED also the effects produced on the Italian coasts by

the few tsunamis generated outside national boundaries, this will be the future task we would like to achieve in the near future and this will also involve the inclusion of tsunamis occurred in the Euro-Mediterranean area in recent years.

## Acknowledgements



Authors thank Dr. Mario Locati, for the technical support during the realization of the ITED database and for contributing to link the ITED database to the INGV Gazetteer and to ASMI and DBMI databases. Authors wish to thank also Dr. Andrea Rovida and Dr. Andrea Antonucci (INGV, Milan) for their contribution to this paper.

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





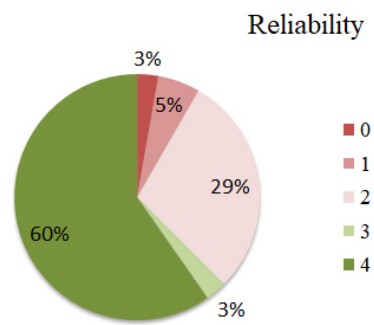

Figure 1: Percentages of Italian tsunamis present in the EMTC according to their reliability.



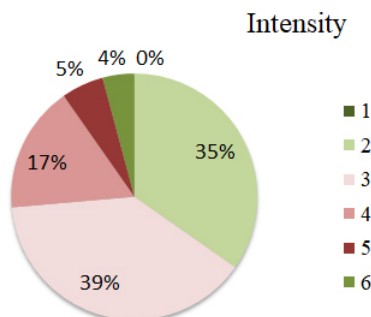

Figure 2: Percentages of Italian tsunamis present in the EMTC according to Sieberg-Ambreseys intensity.



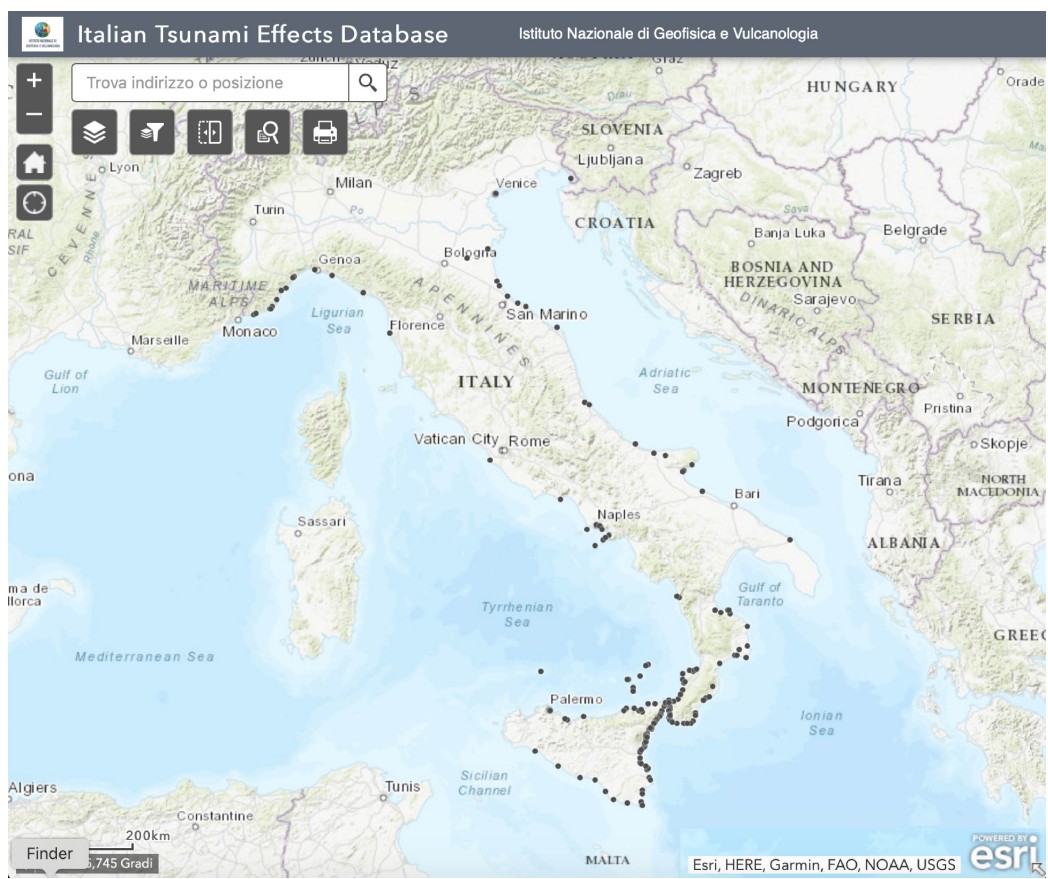

Figure 3 – Main screen of ITED webApp showing the PNs for which tsunami effects were observed, the majority of observations is concentrated in Southern Italy, in particular along Eastern Sicily and

5   Southern Calabria coasts (Service Layer Credits: sources: ©Esri, HERE, Garmin, Intermap, increment P Corp., GEBCO, USGS, FAO, NPS, NOAA; NRCAN, GeoBase, IGN, Kadaster NL, Ordnance Survey, Esri Japan, METI, NASA, Esri China(Hong Kong), ©swisstopo, ©OpenStreetMap contibutors 2019. Distributed under a Creative Commons BY-SA License).





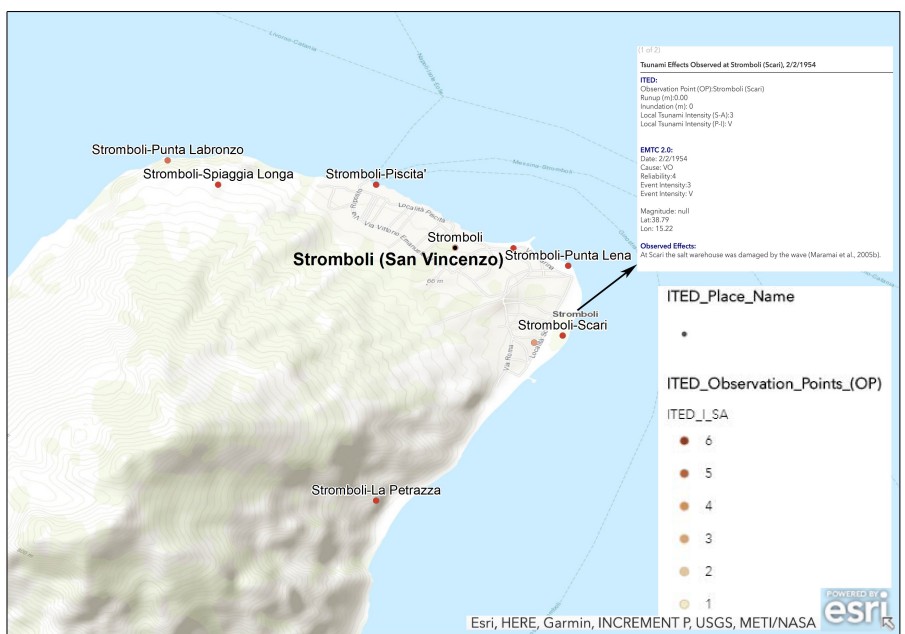

Figure 4 Stromboli Island.-The black dot indicates the PN (Stromboli-San Vincenzo), whereas the brown dots indicate all the nearby OPs, colored according to local tsunami intensity. By clicking on the OP a pop-up allow the user to get details about the tsunami effects at that point (Service Layer Credits: sources: ©Esri, HERE, Garmin, Intermap, increment P Corp., GEBCO, USGS, FAO, NPS, NOAA, NRCAN, GeoBase, IGN, Kadaster NL, Ordnance Survey, Esri Japan, METI, NASA, Esri China(Hong Kong), ©swisstopo, ©OpenStreetMap contibutors 2019. Distributed under a Creative Commons BY-SA License).

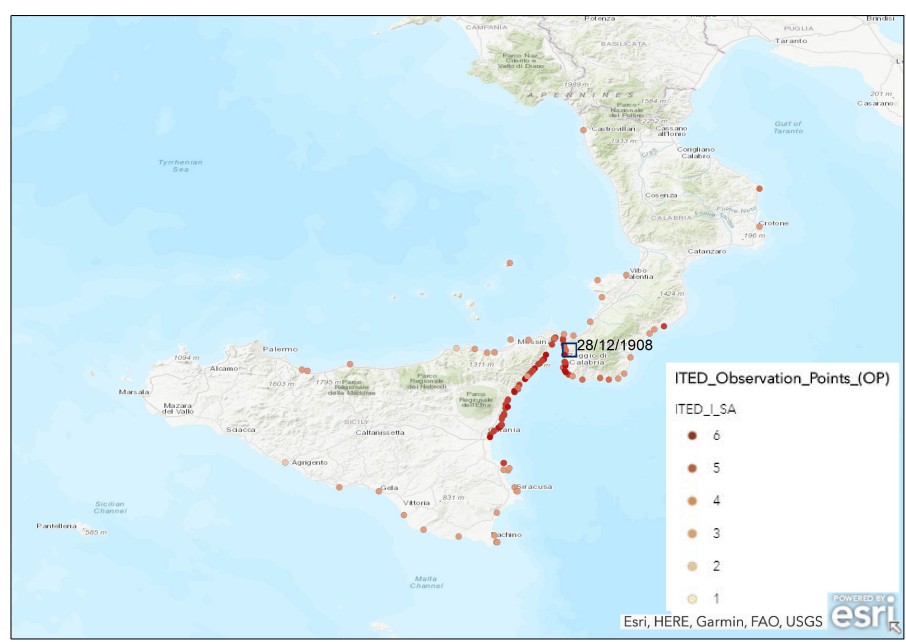

Figure 5 Snapshot of the OPs where the 1908 tsunami was observed, OPs symbols are colored according to the local tsunami intensity (Service Layer Credits: sources: ©Esri, HERE, Garmin, Intermap, increment P Corp., GEBCO, USGS, FAO, NOAA, NPS, NRCAN, GeoBase, IGN, Kadaster NL, Ordnance Survey, Esri Japan, METI, NASA, Esri China (Hong Kong), ©swisstopo, ©OpenStreetMap contibutors 2019. Distributed under a Creative Commons BY-SA License).

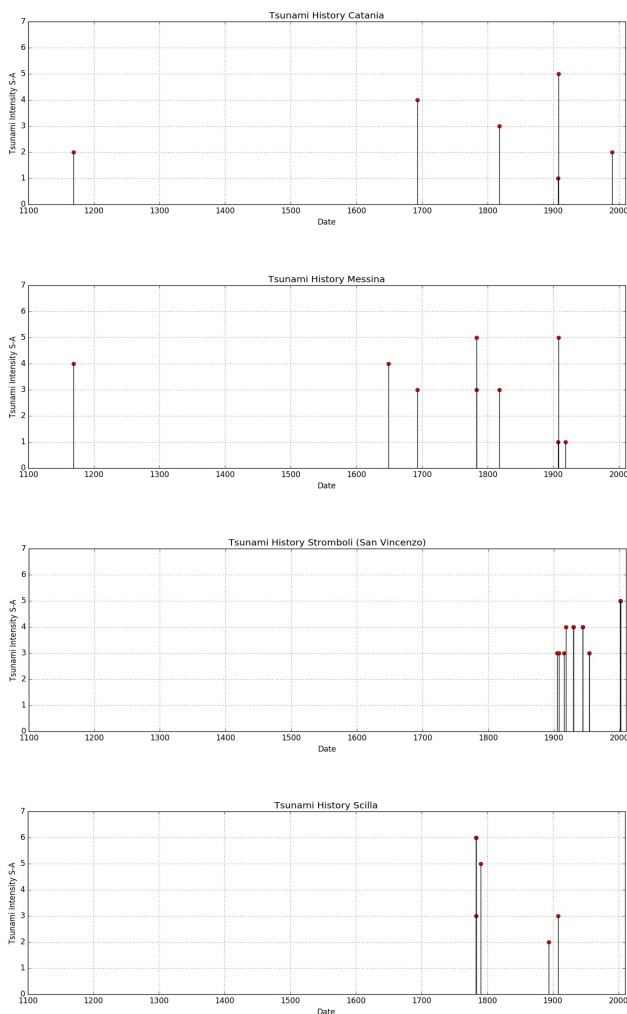

Figure 6 – Tsunami histories contained in ITED for Catania, Messina, Stromboli and Scilla.


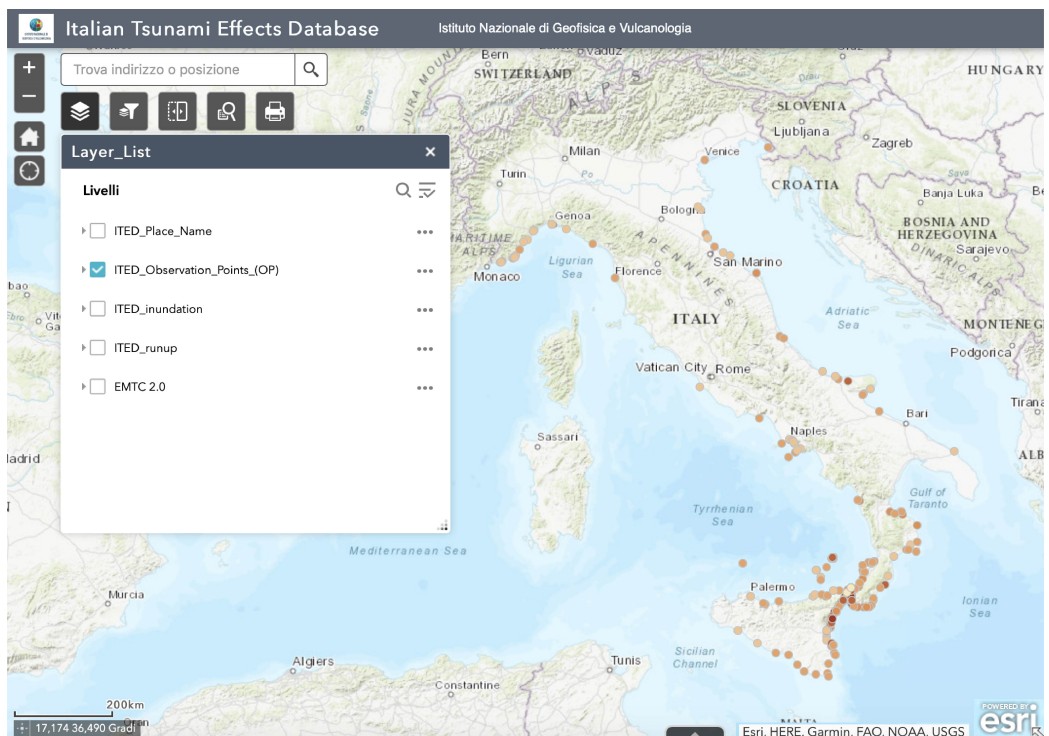

Figure 7 Screenshot of the WebApp layer ITED_Obervation_Point. The complete Layer list is also shown (Service Layer Credits: sources: ©Esri, HERE, Garmin, Intermap, increment P Corp., GEBCO, USGS, FAO, NPS, NRCAN, NOAA, GeoBase, IGN, Kadaster NL, Ordnance Survey, Esri Japan, METI, NASA, Esri China(Hong Kong), ©swisstopo, ©OpenStreetMap contibutors 2019. Distributed under a Creative Commons BY-SA License).

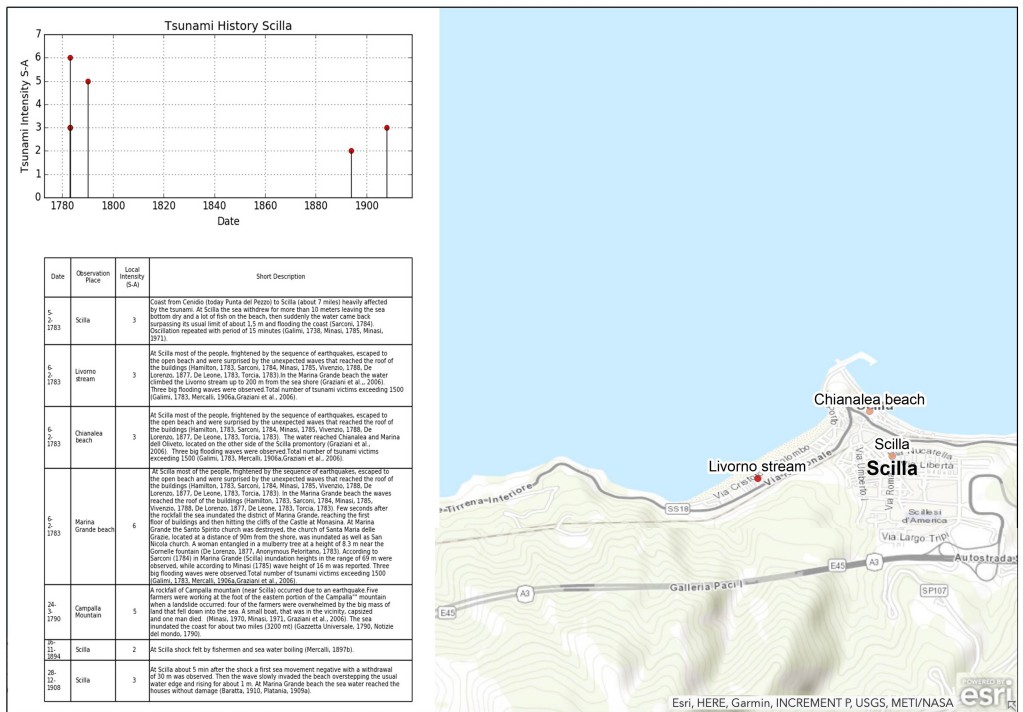

Figure 8. Tsunami history pop-up at PN Scilla (Service Layer Credits: sources: ©Esri, HERE, Garmin, Intermap, increment P Corp., GEBCO, USGS, FAO, NPS, NRCAN, GeoBase, IGN, Kadaster NL, Ordnance Survey, Esri Japan, METI, NASA, Esri China(Hong Kong), ©swisstopo, ©OpenStreetMap contibutors 2019. Distributed under a Creative Commons BY-SA License).


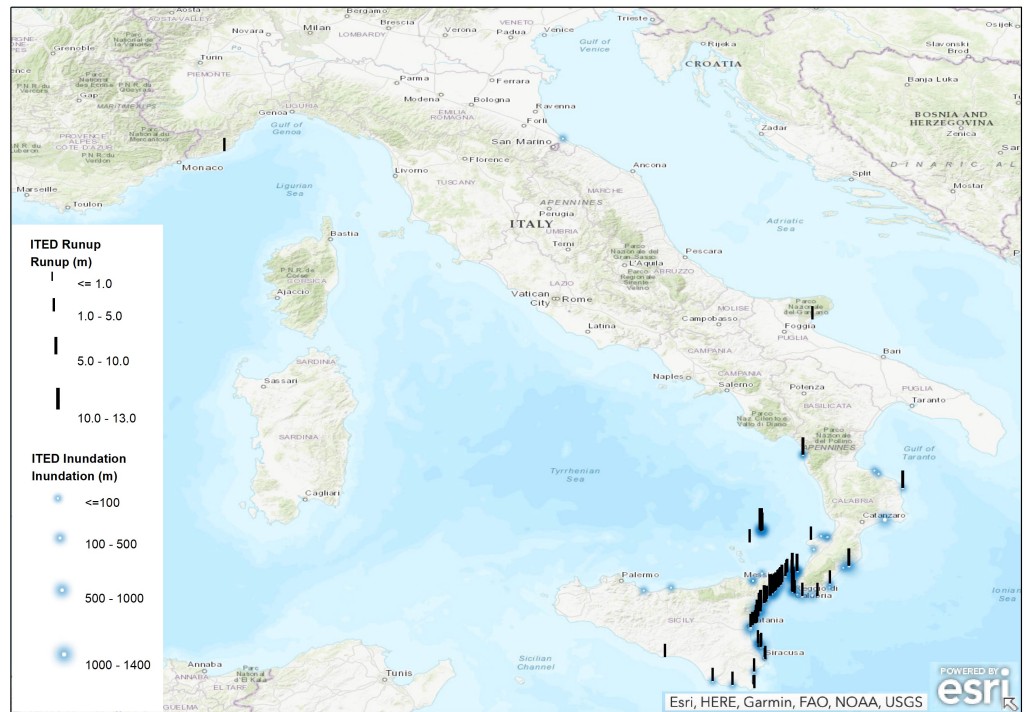

Figure 9 – Screenshot of the WebApp showing ITED_runup (black bars) and ITED_inundation (blue halos) layers; symbols are proportional to the measured values (Service Layer Credits: sources: ©Esri, HERE, Garmin, Intermap, increment P Corp., GEBCO, USGS, FAO, NPS, NRCAN, NOAA, GeoBase, IGN, Kadaster NL, Ordnance Survey, Esri Japan, METI, Esri China(Hong Kong), ©swisstopo, ©OpenStreetMap contibutors 2019. Distributed under a Creative Commons BY-SA License).

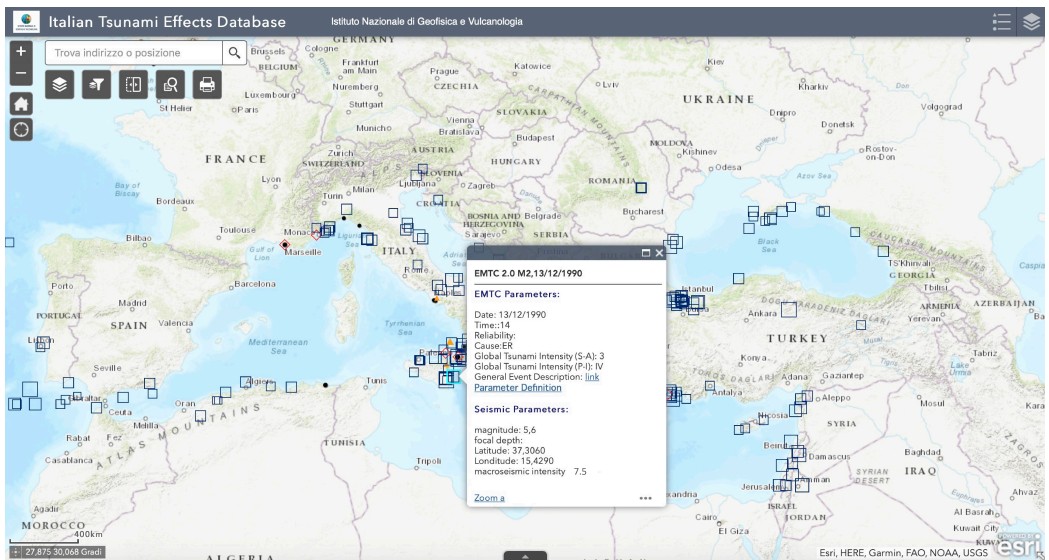

Figure 10 – Screenshot of the EMTC2.0 layer. The pop-up is referred to the 1990 Eastern Sicily tsunami (Service Layer Credits: sources: ©Esri, HERE, Garmin, Intermap, increment P Corp., GEBCO, USGS, FAO, NPS, NRCAN, NOAA, GeoBase, IGN, Kadaster NL, Ordnance Survey, Esri Japan, METI, Esri China(Hong Kong), ©swisstopo, ©OpenStreetMap contibutors 2019. Distributed under a Creative Commons BY-SA License).

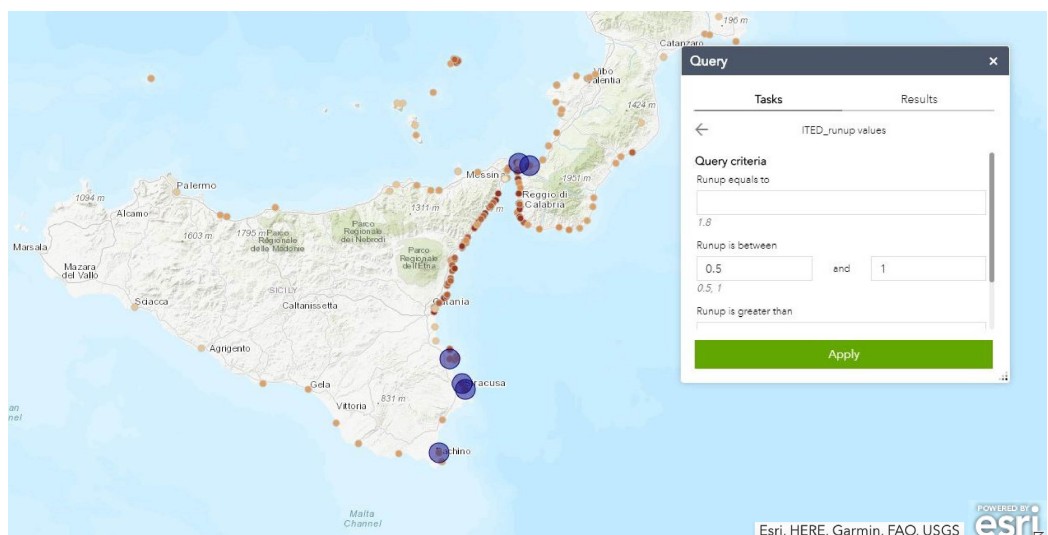

Figure 11 – Screenshot of ITED Webapp showing an example of query by runup value criteria (Service Layer Credits: sources: ©Esri, HERE, Garmin, Intermap, increment P Corp., GEBCO, USGS, FAO, NPS, NRCAN, GeoBase, IGN, Kadaster NL, Ordnance Survey, Esri Japan, METI, Esri China(Hong Kong), ©swisstopo, ©OpenStreetMap contibutors 2019. Distributed under a Creative Commons BY-SA License).



| EMTC.2.0 | N events |
|---|---|
| Source parameters | 53 |
| Generating cause | 6 |
| reliability | 9 |
| Intensity | 38 |
| Origin date | 2 |

Table 1 Number of updated events by parameters in EMTC2.0 respect to EMTC.