# Peer review of "Italian Tsunami Effects Database (ITED): the first database of tsunami effects observed along the Italian coasts"

_Natural Hazards and Earth System Sciences, 2019_

## Referee Comment (RC1) · Efim Pelinovsky (Referee) · 10 Aug 2019

Review on "Italian Tsunami Effects Database (ITED): the first database of tsunami effects observed along the Italian coasts" by Alessandra Maramai, Laura Graziani, Beatriz Brizuela

It is evident that tsunami catalogues and databases are very useful for scientists, decision makers, and citizens. That is why I consider the created Italian tsunami effect database to be an important step in tsunami knowledge dissemination and, therefore, recommend that the reviewed paper should be published.

[Figure]

Creative Commons BY license logo

There are some minor comments:

1. The mentioned link https://tsunamiarchive.ingv.it/ited does not work

2. I have not found which languages are used in the catalogue, I recommend using both - Italian and English, or, it may be published in several languages for tourist proposes.

I recommend publishing this paper as it is.
* * *

---

## Author Comment (AC1) · 15 Aug 2019

Dear Efim, thank you very much for your positive comment. We are very honored that you appreciated our work. About the problem you highlighted in connecting to the webapp, I would like to say that at present the webapp can be accessed through the following website: https://tsunamiarchive.ingv.it/ited.1.0 clicking on the available link at the Data Access section: http://ingv.maps.arcgis.com/apps/webappviewer/index.html?id=a14231712588470ea1c4454301b8294c

Please, let us know if you can access the webapp in this way. thanks Alessandra

---

## Referee Comment (RC2) · Jean Roger (Referee) · 19 Aug 2019

Dear Editor,

Please find below my comments concerning the paper of Maramai et al. entitled "Italian tsunami effects database (ITED): the first database of tsunami effects observed along the Italian coasts".

Most of the available global tsunami catalogues provide tsunami information about the source of the tsunamis (earthquake, landslide, volcanic eruption, etc.), date and time of occurrence, adding, when it is possible, data concerning their maximum run-up height,

period, etc. But most of the time there is a lack of detailed information concerning the different localities where the tsunami has been reported and there is no mean to visualize directly on maps the width of the impacted region.

The paper of Maramai et al. describes a new interesting way to provide tsunami data to end users called ITED (for Italian Tsunami Effects Database). This database linked to the CPT115 seismic database provides 300 tsunami observations at 225 different observation points located along the Italian coastline. In order to represent quickly and easily the different available layers of data (5 layers are available and described within the paper), an easy-to-use WebApp has also been developed; it is also presented in the paper. This WebApp allows the user to visualize and extract data in a specific area : for example, it is possible to ask for the list of events having impacted Messina over the past centuries, each event being assigned an intensity and other information like date, time reliability, etc. In addition, this study allows the authors to add 2 new events to the Italian catalogue of tsunami. The explanation of the methodology applied to prepare the database as well as the use of the WebApp are well explained and sufficient examples are provided to understand everything.

Anyway, I would recommend considering the following comments before the publication of the paper in NHESS.

1) They are three things I would like to access in this database and the first one concerns meteotsunamis. In fact, the Adriatic Sea is well known for this frequent phenomenon (e.g. Orlic, 2015) but other places have also been concerned by the "Marrobbio" and there are several examples on the Italian coast, especially in the South (e.g. Candela et al., 1999 ; Sepic et al., 2018).

2) There is also a lack of data about tsunamis generated outside the Italian area but the authors indicate clearly in the conclusion that the distant sources would be integrated into ITED in the future. Anyway, it would be of great interest to indicate in this paper the percentage of tsunamis triggered by distant sources reported or recorded in Italia

in comparison to local ones. The same way, it would be interesting to have an idea of the amplitudes of those tsunamis because, in hazard mapping, it is of great concern to consider all the tsunamis being able to impact a specific coastline and not only the local ones.

3) Finally, I would like to access to an additional layer within the WebApp locating the tsunami sources linked to the reported tsunamis (with information about the earthquake magnitude for example) and not only the impacted zones.

4) Additional comments:

I have listed below several comments concerning mistakes in the paper: - P. 3, line 29: I would write "fills in" instead of "populates" - P. 3, line 35: change the "-" after "ASMI" → "," - P. 4, line 2: change the "-" after "DMBI" → "," - P. 4, line 24: "7 tsunamis, most of which" → please remove "most of which" as you only have 4 tsunamis left. . . - P. 4, line 27: please correct "The February 6, 1783" - P. 5, line 3: please add "," after "Catania" - P. 5, line 6: please change "Fig. 7" to "Fig. 6" - P. 5, line 12: please correct "January 11, 1693" - P. 5, line 15: add a space between "1" and "m" - P. 6, line 12: remove space after website address - P. 6, line 17: please correct "the metric and iii) the EMTC2.0" - P. 6, line 19: change "the layers available on the screen" - P. 6, line 26: correct "ITED_Place_Name layer" - P. 7, line 4: please correct "which kind of information" - P. 7, line 24: "are exposed to tsunami attack." → remove "attack" - P. 8, add a blank line between first and second references (line 8 and 9)

---

## Referee Comment (RC3) · Anonymous Referee #3 · 20 Aug 2019

This is in general a nice informative paper. I would just like to make a few recommendations to the authors and request the corrections of some typos:

-Page 1, Line 21: Include reference after Sieberg-Ambraseys

-Page 2, Line 7 and Figure 2. The scale of intensity in the figure should be defined, as it has previously been done for reliability. Is 1 the weakest and 4 the strongest, or the other way around? Perhaps it could be defined as labels in the colorbar of Figure 2.

-Page2, Line 26. Please provide reference after "INGV Gazzetter".

-Page 4, Line 29. "...sequence of earthquakes sought on the beach..." should read

something like "...sequence of earthquakes sought shelter on the beach..."?

Page 4, Line 33/ Something should be mentioned about how "runup" is defined to the best of the authors's knowledge. Is the definition proposed by the International Tsunami Information Center in its published Glossary used here? The same for "inundation" and other metrics compiled.

Page5: Line 6. "In Figure 7, the tsunami-histories...." should read "In Figure 6, the tsunami-histories...."

Page 5: Line 26. I believe by "achieved", the authors mean "archived".

Page 6, Line 16 and 17: Should the numerals iiii and iiiii be replaced by iv and v?

---

## Referee Comment (RC4) · Anonymous Referee #4 · 20 Aug 2019

The paper describes structure and the data content of the ITED (Italian Tsunami Effect Database) that was created as a supporting part of the EMTC (Euro Mediterranean Tsunami Catalogue) database. The data on coastal tsunami manifestation, compiled within the ITED, have been collected from a variety of published and archival sources and were carefully revised. They are extremely important for assessment of tsunami hazard along the Italian coast. The paper is written in a compact and informative manner and can be published as it is, after several miner corrections listed below.

Page 4 line 4 Reference for (Guidibini et al., 2018) is absent in the Reference list.

Page 4 line 33 ". . . where inundation values are specified,. . ." . " values" is a very

general term, it should be replaced with more specific term like "extent" or "distance"

Page 5 line 35 Reference for (Guidibini et al., 2018) is absent in the Reference list

Page 6 line 13 The indicated web-reference does not work

Page 6 line 14 "the OP" should be types as "OPs"

Page 8 line 35 (References) Is the paper (Platania, 1909) written in Italian? In this case "(in Italian)" should be added to its reference

Page 9 line 1 (References) The reference Rovida A., Locati M., Antonucci A., Camassi R. (eds.): Italian Archive of Historical Earthquake Data (ASMI). Istituto Nazionale di Geofisica e Vulcanologia (INGV). https://emidius.mi.ingv.it/ASMI/ in the text is cited in two different ways Rovida et al., 2016 (page 3 line 32) Rovida et al., Eds (page 3 line 35) Rovida et al., 2016 (page 5 line 35)

Page 9 line 12-13 (References) The reference TSUMAPS-NEAM Interactive Tool: http://ai2lab.org/tsumapsneam/interactive-hazard-curve-tool/ last access July 2019. is not mentioned in the text

Page 12 line 3 (in caption to Figure 3) webApp should be typed as WebApp (as in the text)

In some places, the English should be polished up (several examples are below): "being affected during times" (page 5 line 4) "accordingly to the new achieved info" (page 5 line 27) "About the 1690 event, . . ." (page 5 line 35) "provoking victims" (page 5 line 36) "respect to EMTC" (page 6 line 4) "struck localities" (page 6 line 15) "The events. . . are divided into 4 main causes" (page 6 line 35) "filtering expressions" (page 7 line 8) "also distant sources, including in ITED also the effects produced on the Italian coasts" (page 7 line 29)

Please also note the supplement to this comment:
https://www.nat-hazards-earth-syst-sci-discuss.net/nhess-2019-241/nhess-2019-241-

RC4-supplement.pdf

---

## Author Comment (AC2) · 21 Aug 2019

Dear Jean Roger

Thank you for your valuable suggestions, we will try to take them into account as much as possible.

Regarding the specific comments on meteotsunamis we are aware that metoetsunamis have been frequently observed along the Italian coasts and that they play a key role in the hazard assessment along the Italian coasts. This is why we are starting to consider to build a new WebApp specifically devoted to the observations of this this phenomenon

along the Italian coasts. Nevertheless ITED was created to better exploit data concerning Italian tsunamis contained in EMTC, a catalogue of the tsunamis occurred in the Euro-Mediterranen region, where meteotsunamis were not taken into account. We believe that inserting a new category of events only for the Italian part would result in a non homogeneous product.

Adding observations of tsunamis effects caused by distant sources, as we specified in the paper, it's a task we would like to achieve in the near future as well as the inclusion of tsunamis occurred in the Euro-Mediterranean area in recent years. We already know that the number of events of this kind is very small, counting for about a few units, which may reach barely the 4% of the Italian events. The only reliable data of tsunami observation (measures) along the Italian coasts caused by distant tsunami is the 2003 Boumerdes event that was recorded in Genoa with amplitudes that hardly exceeded 0.05-0.08 m. The lack of such kind of information is mostly due to the lack of instrumental data that could have recorded small amplitude variations. As regards historical events ancient sources testify that the destructive 365 Crete event was observed along Sicily coasts but the description is too vague to identify a specific OP.

As far as concern the WebApp and possibility to isolate the information regarding each source and the effects related to it, we agree that it would be interesting. We actually tried to create a direct filter to produce such an information, but due to the database architecture and to some limitations of the ESRI WebApp builder we did not manage to set it. We are trying to come out with a new layer that in a way could show what you asked, nevertheless, at present there is a way to obtain the information you are interested in without adding a new layer, this can be done starting from the ID value of each event of EMTC ("EMTC_id").

The procedure to follow is: a) First of all, switch on the EMTC 2.0 layer and click the event you are interested in, when the popup appears click on the "three dot" on the lower right corner and select "View in Attribute Table" See figure 1.

b) On the Attribute Table search for the column named "EMTC_id" and copy the value (i.e. 212)

c) On the Attribute table, select "Options" on the left upper corner, then "Filter", "Add Expression", then set the expression as "EMTC_ID" is and paste the EMTC_id value that you copied on (b). Figure 2.

d) You should repeat the procedure for the other layers that you would like to filter. Switch the "ITED Observation Points (OP)", Open the Attribute Table (by clicking the "three dot" symbol next to the layer). On the Attribute Table select "Options" on the left upper corner, then "Filter" and "Add Expression", select the "EMTC_ID" is and paste the EMTC_id value that you copied on (b) The result should be a figure like figure 3.

As we said before, we are trying to find a more straight forward way to let the user export the this kind of information, in case we don't have any luck, we could insert the above mentioned procedure in the paper.
* * *
[Figure]

**Fig. 1.**

Filter ✕

✚ Add expression   ✚ Add set

Display features in the layer that match the following expression.

| EMTC_id (Number) | ▾ | is | ▾ | 212 | ⚙ ✕ |

OK   Cancel

**Fig. 2.**

[Figure]

**Fig. 3.**

---

## Author Comment (AC3) · 21 Aug 2019

Thank you very much for your useful recommendations; we will make the necessary revisions to meet your requirements as well as include the definition of the used metrics referring to ITIC Glossary.

---

## Author Comment (AC4) · 21 Aug 2019

Thank you very much for your useful comments that we will follow in the best possible way including the revision of English. As far as the link to the WebApp is concerned we realized that there is a mistake in the link, the correct link is

https://tsunamiarchive.ingv.it/ited.1.0/

---

## Referee Comment (RC5) · Anonymous Referee #5 · 3 Sep 2019

The paper describes a new database of tsunami effects observed in Italy, which is accessible through a web application. The database will be useful for future tsunami hazard evaluations, and the paper is thus of interest to the wider tsunami and natural hazards communities, also outside Italy.

I recommend publishing the paper, but before publication I recommend the following revisions:

- I would have liked to see a more clear description of the possibilities this dataset opens for the scientific community. What can we do with this dataset that we could not do before?

[Figure]

- Except for a small comment on the tsunami history at Stromboli, there is no discussion of the level of completeness of the dataset. This is of huge importance when using the data for tsunami hazard assessment, and should be addressed.

- I recommend restructuring the paper with separate sections focusing on "methods" and "results". This would make it easier for the reader to see what has been done to collect the data, and it would prevent a lot of repetition, which is in the current version.

- At the same time, I would have liked to see more detailed descriptions of the strategies for data collection and interpretation, and of the changes made to the EMTC. Ideally, changes made to EMTC could be included in an online supplement, which would make it easier for users of the previous version to correct their work.

- In my view, the description of the database and the web application could be shortened somewhat, and 1-2 of the example plots could be left out. I would expect any reader to go directly to the database to have a look at the functionalities, so figure examples do not need to be included for all functionalities.

- Figure 1: The meanings of the different reliability levels (or criteria for assessment) should be provided.

- Figure 2: The color scale is not logical, I recommend "shifting" the colors such that I=6 is red instead of green.

- It would be useful to include a short description of the intensity scales used.

- The manuscript contains many small grammatical errors and wrong wording, which should be fixed. For example, on page 1, line 14, I assume "interested" should rather be "affected"? Page 7, line 27: public awareness will not reduce tsunami hazard, but may contribute to reducing tsunami risk.

- At several places, the authors state that the database focuses on propagation effects rather than the tsunami source. In my understanding, "propagation" relates to the propagation of the tsunami in the deep sea, whereas the effect on the coast is better

covered by the term "inundation".

---

## Author Comment (AC6) · 19 Sep 2019

Dear Referee, Thank you for your useful recommendations, we will try to take them into account when revising the paper in the meanwhile we would like to reply to your suggestions

- I would have liked to see a more clear description of the possibilities this dataset opens for the scientific community. What can we do with this dataset that we could not do before?

ITED can provide more easily the information contained in EMTC, first of all it allows

querying the database through different keys moreover the user can for example re-
trieve information on the coastal sites (PN) in order to have an idea on how it was
interested by tsunamis in the past or can know the extent of the effects of a single
tsunami.

- Except for a small comment on the tsunami history at Stromboli, there is no discussion
of the level of completeness of the dataset. This is of huge importance when using the
data for tsunami hazard assessment, and should be addressed.

The number of tsunamis contained in ITED is quite low and therefore statistical anal-
yses performed with those numbers would lead to non-robust results. ITED database
spans a time interval of two millenniums but t is largely incomplete in the first where
only the 79 AD. Vesuvius eruption tsunami is included. Considering the frequency of
events related to reliability only in the last three centuries the number of entries in the
catalogue begin to be stable suggesting the achievement of a certain level of complete-
ness.

- At the same time, I would have liked to see more detailed descriptions of the strategies
for data collection and interpretation, and of the changes made to the EMTC. Ideally,
changes made to EMTC could be included in an online supplement, which would make
it easier for users of the previous version to correct their work.

Regarding the descriptions of the strategies of data collection and interpretation we
would like to make clear that in ITED we made an update of the Italian section of
EMTC following the criteria used during the first release. We are working currently on
the collection of data regarding tsunamis occurred from 2014 to present in order to
updated the EMTC, in this frame we will provide all the details of the changes made to
the previous version.

-I recommend restructuring the paper with separate sections focusing on "methods"
and "results". This would make it easier for the reader to see what has been done to
collect the data, and it would prevent a lot of repetition, which is in the current version.

-in my view, the description of the database and the web application could be shortened somewhat, and 1-2 of the example plots could be left out. I would expect any reader to go directly to the database to have a look at the functionalities, so figure examples do not need to be included for all functionalities.

We will try to modify following your recommendations.

―――――――――――――――